# Identification and Characterization of p300-Mediated Lysine Residues in Cardiac SERCA2a

**DOI:** 10.3390/ijms24043502

**Published:** 2023-02-09

**Authors:** Przemek A. Gorski, Ahyoung Lee, Philyoung Lee, Jae Gyun Oh, Peter Vangheluwe, Kiyotake Ishikawa, Roger Hajjar, Changwon Kho

**Affiliations:** 1Cardiovascular Research Institute, Icahn School of Medicine at Mount Sinai, New York, NY 10029, USA; 2Research Institute for Korean Medicine, Pusan National University, Yangsan 50612, Republic of Korea; 3New Drug Development Center, Osong Medical Innovation Fundation, Osong, Seoul 02841, Republic of Korea; 4Department of Cellular and Molecular Medicine, KU Leuven, 3000 Leuven, Belgium; 5Phospholamban Foundation, 1775 ZH Amsterdam, The Netherlands; 6Division of Applied Medicine, School of Korean Medicine, Pusan National University, Yangsan 50612, Republic of Korea

**Keywords:** calcium ATPase, cardiac muscle, acetyltransferase, post-translational modifications

## Abstract

Impaired calcium uptake resulting from reduced expression and activity of the cardiac sarco-endoplasmic reticulum Ca^2+^ ATPase (SERCA2a) is a hallmark of heart failure (HF). Recently, new mechanisms of SERCA2a regulation, including post-translational modifications (PTMs), have emerged. Our latest analysis of SERCA2a PTMs has identified lysine acetylation as another PTM which might play a significant role in regulating SERCA2a activity. SERCA2a is acetylated, and that acetylation is more prominent in failing human hearts. In this study, we confirmed that p300 interacts with and acetylates SERCA2a in cardiac tissues. Several lysine residues in SERCA2a modulated by p300 were identified using in vitro acetylation assay. Analysis of in vitro acetylated SERCA2a revealed several lysine residues in SERCA2a susceptible to acetylation by p300. Among them, SERCA2a Lys514 (K514) was confirmed to be essential for SERCA2a activity and stability using an acetylated mimicking mutant. Finally, the reintroduction of an acetyl-mimicking mutant of SERCA2a (K514Q) into SERCA2 knockout cardiomyocytes resulted in deteriorated cardiomyocyte function. Taken together, our data demonstrated that p300-mediated acetylation of SERCA2a is a critical PTM that decreases the pump’s function and contributes to cardiac impairment in HF. SERCA2a acetylation can be targeted for therapeutic aims for the treatment of HF.

## 1. Introduction

Heart failure (HF) is a pandemic that affects about 64.3 million people worldwide, and its incidence and prevalence are correlated with aging [1,2]. The reduced expression and activity of the cardiac sarco-endoplasmic reticulum Ca^2+/−^ATPase (SERCA2a) pump are observed in the failing human heart, resulting in a decreased rate of relaxation and force of contraction [3]. Reintroduction of the SERCA2a gene by viral vector-mediated gene transfer in various animal models of HF has been showing significant improvements in cardiac function and survival to be sufficient to confirm SERCA2a as an effective therapeutic target for HF [4,5,6]. The Ca^2+^ sequestering activity of SERCA2a is mainly regulated by the endogenous protein phospholamban (PLN), which lowers its apparent affinity for Ca^2+^ [7]. Other mechanisms of SERCA2a regulation, such as microRNA or hormonal regulation of gene expression and post-translational modifications (PTMs) of the SERCA2a protein, have also been linked to HF [8,9,10]. Together, these mechanisms emphasize the central role of SERCA2a in cardiac function and the complexity of the regulatory mechanisms involved in maintaining the proper function of the Ca^2+^ pump.

Protein PTMs have multifaceted roles and are rapidly being recognized as important mechanisms of key protein regulation in hearts. PTMs have been known to alter protein function by creating new protein binding sites, abrogating protein-protein interactions, or inducing structural change by the allosteric effect. More importantly, the cross-talk between different PTMs is an emerging paradigm in biology [11,12]. Over the years, several PTMs of SERCA2a, such as glycosylation, glutathionylation, and nitration, have been reported to have a direct implication in modulating its activity under normal and stress conditions [13,14,15]. In the previous study, we identified a new PTM, SUMOylation, that plays a critical role in the SERCA2a function [16]. The small ubiquitin-like modifier 1 (SUMO1), through its covalent attachment to lysines 480 and 585 in SERCA2a, has been shown to enhance protein stability and activity of SERCA2a. Cardiac-specific overexpression of SUMO1 by adeno-associated virus (AAV)-mediated gene delivery significantly improved cardiac function and protected SERCA2a from pathological stresses in mouse and porcine models of HF [16,17,18]. Additionally, we showed that SUMOylation of SERCA2a could be upregulated by activating the activity of a SUMO1-activating enzyme (E1) using specific pharmacological molecules, resulting in improved contractile function and further supporting the cardioprotective properties of SUMOylation in the heart [19].

Acetylation is one of the most common PTMs and has a profound effect on protein function [20]. The process of protein acetylation is enzymatically mediated by lysine acetyltransferases (KATs) and can be reversed by specific lysine deacetylases (KDACs). During the last decade, several KATs and KDACs have been shown to play a pivotal role in cardiac development and pathological processes of cardiac remodeling. For instance, cardiac-specific expression of p300, a ubiquitously expressed transcriptional cofactor and an acetyltransferase, has been shown to cause hypertrophy and HF in adult mice [21,22]. On the other hand, Sirtuins (silent mating type information regulation 2 homolog, HDAC class III) is a critical molecule in ischemia/reperfusion studies due to their impact on aging, apoptosis, metabolic homeostasis, and stress responses [23,24]. Despite the increasing number of studies related to protein acetylation/deacetylation, our understanding of the exact role of acetyltransferases and deacetylases in cardiac function is incomplete, as these enzymes have numerous protein targets.

Previously, we reported that SERCA2a is acetylated and that this acetylation is more prominent in the setting of HF [25]. Moreover, we showed that SERCA2a is acetylated by p300 and is deacetylated by SIRT1 (Sirtuin 1). Additionally, we confirmed that a decreased SERCA2a acetylation by pharmacological activation of SIRT1 in HF restored the SERCA2a function and improved cardiac function. This mechanism of action is related to the lysine 492 residue on SERCA2a, which is part of a group of residues forming the ATP-binding pocket. However, the role of p300 as acetyltransferases of SERCA2a acetylation is still unknown. Herein, our data provide clear evidence that SERCA2a is a direct substrate of p300. p300 significantly increased acetylation and inhibited the SERCA2a function. Additionally, we demonstrated that p300 regulates SERCA2a acetylation at multiple lysine residues. This study provides new insight into the regulation mechanism of SERCA2a by PTMs.

## 2. Results

### 2.1. Increase of p300 Is More Prominent in Human Heart Failure

In our analysis of SERCA2a PTMs in human myocardium, we previously reported that SERCA2a is acetylated and that this acetylation is more prominent in failing human hearts [25]. This increase in SERCA2a acetylation was also observed in cardiac tissue from transverse aortic constriction (TAC)-operated mice, implying that an increase in acetylation is a general mechanism underlying the impairment of SERCA2a function in failing hearts. Based on these observations, we determined that the expression levels of key acetyltransferases were elevated in failing human hearts (Figure 1). Especially, a significant increase in p300 protein and mRNA levels was observed in failing human hearts compared to other acetyltransferases, such as PCAF (p300/CBP-associated factor) and GCN5 (general control of amino-acid synthesis protein 5) (Figure 1B,C). Taken together, these results suggest that p300 appears to be a strong candidate for mediating the acetylation of SERCA2a.

### 2.2. p300 Directly Interacts with and Acetylates SERCA2a Resulting in Diminished Calcium Transport Activity

We set out to screen the most common cytosolic acetyltransferases for their acetylation of SERCA2a. In HEK293 cells, SERCA2a was co-expressed with p300, CBP (CREB binding protein), PCAF or GCN5, after which immunoprecipitation (IP) was performed to determine the role of acetyltransferases in the acetylation of SERCA2a. p300 was the only acetyltransferase to significantly increase acetylation levels of SERCA2a and strongly interact with SERCA2a (Figure 2A,B). Additionally, the endogenous interaction between p300 and SERCA2a was verified in porcine and human hearts (Figure 2C,D).

The interaction between p300 and SERC2a was confirmed in cell-based experiments. In HEK293 cells, SERCA2a was co-expressed with different p300 dosages, after which IP and ATPase activity assays were performed. The results indicated that the levels of SERCA2a acetylation and the extent to which the pump interacts with p300 are more prominent as the concentration of p300 increases (Figure 3A). To further confirm the acetylation of SERCA2a by p300 in vitro, SERCA2a purified from normal porcine hearts was incubated in the presence or absence of recombinant human p300 (rhp300). Immunoblotting with anti-acetyl-lysine or anti-SERCA2a antibodies revealed that SERCA2a is acetylated, and its acetylation is increased along with increased concentrations of p300 (Figure 3B). Furthermore, SERCA2a co-expressed with p300 exhibited significantly decreased ATPase activity as compared to SERCA2a alone (Figure 3C).

Next, the functional effects of SERCA2a acetylation mediated by p300 were determined using cardiomyocytes. Adult cardiomyocytes (ACMs) isolated from normal mice were treated with CTPB (p300 activator) or L002 (p300 inhibitor) to regulate SERCA2a acetylation mediated by p300. It was confirmed that both compounds modulate the acetylation status of SERCA2a in ACMs as well as in vitro assays (Appendix A). SERCA2a acetylation was increased by treatment with CTPB, while its acetylation was remarkably decreased after treatment with L002. Either CTPB or L002 treatment did not alter SERCA2a protein expression levels in ACMs. Ca^2+^ transient profiles and mechanical properties of ACMs were measured using a video-based edge-detection system (Figure 3D). The CTPB-treated ACMs exhibited significantly reduced Ca^2+^ handling as demonstrated by decreased Ca^2+^ transient amplitude and increased tau, and reduced contractility as demonstrated by decreased cell shortening and maximal rates of contraction and relaxation in comparison to DMSO-treated ACMs. However, the Ca^2+^ handling and contractility in L002-treated normal ACMs were not changed. Taken together, our data indicate that p300-mediated SERCA2a acetylation affects the mechanical properties of ACMs by regulating SERCA2a activities.

### 2.3. p300 Acetylates SERCA2a at Multiple Lysine Residues

Lysine sites on SERCA2a that are acetylated by p300 were identified by liquid chromatography with tandem mass spectrometry (LC-MS/MS). LC-MS/MS analyses were performed on purified porcine SERCA2a incubated with or without rhp300, in which SERCA2a was either acetylated or not (Table 1 and Figure 4A). A total of 12 putatively acetylated lysine sites were detected. Six putative acetylated lysine sites were in the nucleotide-binding domain; 3 were in the actuator domain; 1 was in the phosphorylation domain; 2 were in the transmembrane domain (Figure 4B). These acetylated lysine sites were highly conserved among human SERCA1a and human SERCA2a. Especially, 6 acetylated lysine sites (e.g., K 328, 329, 492, 510, 514, and 712) appeared to be acetylated in repeated MS analysis, suggesting potential acetylation at these sites (Figure 4C).

The functional significance of the identified 6 acetylated lysine sites of SERCA2a was investigated using acetylation-mimicking mutants of SERCA2a. In order to mimic acetylation, the lysine residue of SERCA2a was exchanged for glutamine (Q). HEK293 cells were transfected with plasmids expressing SERCA2a mutants that mimicked acetylation. The protein levels of K510Q, K514Q, and K712Q SERCA2a were slightly lower than the wild-type SERCA2a (WT SERCA2a) level (Figure 4D). The ATPase activity of these acetylation-mimicking mutants was measured, and the values were normalized by each protein level. Among them, K328Q, K492Q, and K514Q SERCA2a exhibited > 45% reduced ATPase activities compared to WT SERCA2a (Figure 4D and Table 2). These lysine residues are highly conserved in amino acid sequence in SERCA1a, another isoform of SERCA. In particular, K492 and K514 are considered essential for SERCA2a in terms of structural aspects related to enzyme activity as well as actual ATPase activity results. (Figure 4B,E). Taken together, these results suggest that acetylation at K492 and K514 may be associated with the reduced SERCA2a activity, and its acetylation may be mediated by p300.

### 2.4. Acetylation on Lys514 Has a Profound Effect on SERCA2a Function and Stability 

To gain a deeper understanding of the role of the newly identified acetylation sites on SERCA2a and how they might affect its function, we constructed a structural model of SERCA2a using an x-ray crystal structure of rabbit SERCA1a complexed with ATP. The deduced model suggested that acetylation at K514 might alter the ATP-binding pocket to render the ATP-binding of SERCA2a less favorable (Figure 5A). To explore the functional importance of acetylation of SERCA2a at K514, we generated Flag-tagged acetylation mimicking (K514Q) and non-acetylated (K514R) mutants of SERCA2a. The protein levels of K514Q and K514R SERCA2a were slightly lower than the WT SERCA2a level. Though the ATPase activity was normalized by each protein level, K514Q SERCA2a exhibited significantly decreased ATPase activity compared to WT SERCA2a. Interestingly, K514R SERCA2a also showed significantly reduced ATPase activity (Figure 5B).

The estimated half-life of the K514Q SERCA2a was significantly reduced compared to WT SERCA2a. Interestingly, not all acetyl-mimicking lysine mutants of SERCA2a showed a reduced half-life of the protein (Figure 5C). The estimated half-life of K492Q was shown as an example. This result displayed a high correlation with ubiquitin-dependent degradation. HEK293 cells underwent transfection with plasmids expressing WT SERCA2a or K514Q SERCA2a in the presence or absence of ubiquitin. K514Q SERCA2a showed significantly increased levels of polyubiquitin-conjugated SERCA2a protein compared to WT and K492Q SERCA2a (identified acetylated lysine site) (Figure 5D). Taken together, these results suggest that acetylation at K514 might negatively influence the ATPase activity and stability of SERCA2a.

### 2.5. Acetylation on Lys514 Impairs Cardiomyocyte Function 

To further confirm the functional consequences of acetylation of SERCA2a at K514 on contractile function in vitro, adult cardiomyocytes (ACMs) isolated from the conditional cardiac-specific *Serca2* heterozygous knockout mice (SERCA2^+/−^) were infected with adenoviruses expressing WT or K514Q SERCA2a. Conditional cardiac-specific *Serca2* knockout mice have been characterized in previous studies [26]. In the SERCA2^+/−^ ACMs, the native SERCA2a levels were reduced to ~50% of the normal level in ACMs isolated from wild-type mice (WT), and adenovirus-mediated reintroduction of SERCA2a restored the expression level of SERCA2a to ~70% in the SERCA2^+/−^ ACMs (Figure 6A). WT SERCA2a significantly restored the defective Ca^2+^ transient and contractility of SERCA2^+/−^ ACMs. However, the expression of K514Q SERCA2a had no effect on these properties in SERCA2^+/−^ ACMs. (Figure 6B,C). Taken together, these results suggest that the acetylation of SERCA2a at K514 profoundly impairs the SERCA2a activities.

## 3. Discussion

In this study, we confirmed that p300 directly interacts with SERCA2a and acetylates its multiple lysine residues. We observed that protein and mRNA levels of p300 are significantly increased in human HF. A total of 12 putative lysines of SERCA2a acetylated by p300 were identified from in vitro acetylation assay combined with LC-MS/MS analysis. Through biochemical characterization of the identified acetylated lysine sites of SERCA2a using acetyl-mimicking mutants, we found that K514 is critical in modulating SERCA2a activity. This finding was further corroborated by mechanical characterization after reintroducing the K514Q mutant of SERCA2a in SERCA2a deficient ACMs. These experiments clearly demonstrated that the K514 acetylation of SERCA2a by p300 hinders normal calcium transients and cardiomyocyte contractility. 

The more than 300 currently identified PTMs provide great scope for subtle or dramatic alteration of protein structure and function. PTMs modulate a protein’s activity or intracellular localization of substrates by blocking a binding site, creating an additional binding site, or conformational change. It leads to changes in multiple physiological functions. PTMs are predominantly triggered by enzymes, and the enzymes responsible are thus attractive targets for therapeutic interventions. The heart has a high metabolic rate requiring rapid adaptation to new environments and is also sensitive to oxidative stress. In this context, it is not surprising that many commonly observed PTMs (e.g., phosphorylation, glycosylation, lysine acetylation) play critical impacts on cardiovascular diseases.

Protein lysine residues undergo multiple PTMs, such as ubiquitination, SUMOylation, acetylation, methylation, and glycation, suggesting that cross-talk between different PTMs of lysine resides affects SERCA2a’s function in the heart. Previously, we discovered that SUMO1 conjugates to SERCA2a, rendering it more stable and increasing its activity [16]. The therapeutic effect of strategies to enhance SERCA2a SUMOylation through SUMO1 overexpression or small molecule activators has been further validated in pre-clinical animal models of HF [16,17,18,19]. Acetylation is another essential PTM that modulates the activity of SERCA2a. We provided clear evidence that SERCA2a is a direct substrate of SIRT1, a deacetylase. SIRT1 activation significantly reduced acetylation and restored the SERCA2a function, resulting in beneficial outcomes under cardiac insults [25].

Recent studies show the fundamental role of protein acetylation/deacetylation in the regulation of cardiovascular diseases such as hypertension, pulmonary hypertension, diabetic cardiomyopathy, coronary artery disease, arrhythmia, cardiac fibrosis, and heart failure [27]. Protein acetylation/deacetylation is enzymatically regulated by either KATs or KDACs. Although acetyltransferases are considered to be present mainly in the nucleus, an increasing number of studies show that they can shuttle between the nucleus and cytoplasm [28,29]. The most extensively studied KATs in muscle are p300 and its homolog CBP, which play critical roles in the physiological and pathological pathway of cardiomyocytes. For example, p300 functions as a co-activator of the key transcription factor of cardiac remodeling, such as GATA-4 (GATA binding protein 4) [30] and MEF-2D (myocyte enhancer factor-2D) [31]. Several studies suggest that p300 plays an essential role in the growth of cardiac myocytes during development, demonstrated by the phenotype of p300 knockout mice, whereas p300 overexpression promotes symptoms representative of HF, demonstrated by the phenotype of p300 transgenic mice [21,22]. Acetyltransferase p300 is also an essential epigenetic regulator of fibrogenesis. It has been demonstrated that p300 stimulates the synthesis of type I collagen, a key matrix protein that contributes to fibrogenesis, and requires acetyltransferase activity [32]. The importance of p300 function in HF was further validated in animal models of hypertension and MI [33,34]. Pharmacological inhibition of acetyltransferase activity of p300 improved structural abnormalities and cardiac dysfunction induced by cardiac injury [35,36]. Taken together, these findings imply that, in some cases, they may utilize a therapeutic strategy through the regulation of p300 for HF treatment. Both previous and this study showed that p300 expression increases in failing hearts, which induces SERCA2a acetylation. However, upstream regulators and detailed control mechanisms of p300 expression in failing hearts are uncertain.

We identified 12 possible lysine acetylation sites in the SERCA2a protein mediated by p300. Among them, acetylation of K510 and K533 was reported in normal guinea pigs [37]. Eight residues (K128, K169, K205, K476, K492, K514, K533, and K712) were previously reported or newly identified in the SERCA1a and SERCA2a studies isolated from normal pig hearts [38]. K492 is a substrate for SIRT1 deacetylation [25]. Each of the putative acetylating lysines can affect diverse aspects of SERCA2a function. K514 is part of a group of residues forming the ATP-binding pocket of SERCA2a. It has already been reported to play an important role in ensuring proper ATP binding in SERCA proteins [39]. Interestingly, two acetylated lysine residues such as K328 and K329, were identified in the putative binding pocket of SERCA2a that binds with PLN. K328 has already been reported to cross-link with Q23 and K27 of PLN [40,41]. Through IP experiments in our overexpression systems, it was observed that the acetylation-mimic mutant of K328 SERCA2a increases its affinity for PLN by ~2.5-fold compared to wild-type SERCA2a. In-depth studies will be needed to clarify the role of acetylation in the SERCA2a-PLN complex. In conclusion, K514 is highly vulnerable to modification (i.e., acetylation) or substitution, and any change in this site may significantly influence the ATP-binding capacity of SERCA2a less favorably. Interestingly, K514 is possible to affect the stability of SERCA2a, in part through the regulation of ubiquitination.

p300 may affect other molecules involved in heart functions, including other Ca^2+^ handling proteins and contractile machinery. In addition, the in silico acetylation site analysis using the GPS-PAIL tool [42] suggests that the RyR2 Ca^2+^ channel protein, sodium-calcium exchanger, and troponin I contain many acetylation sites. Also, some of these proteins undergo other PTMs, such as phosphorylation; thus, it cannot be ruled out that they could be direct or indirect targets of p300.

SERCA2a is a promising molecular target for HF therapy. PLN is a well-known regulatory protein of SERCA2a, and its reduced phosphorylation contributes to impaired SERCA2a function in failing hearts. Many efforts to increase PLN phosphorylation have led to the development of the continuously active form of protein phosphatase inhibitor I1 (I1c). I1c gene therapy recently initiated a Phase 1 clinical study (NCT04179643). A recent study revealed another protein DWORF which functions as an activator of SERCA2a via replacing PLN [43].

Several groups, including us, have shown that PTMs are important regulatory mechanisms of SERCA2a activity. SUMOylation [16] and glutathionylation [14] function as SERCA2a stimulators, whereas oxidation [44] and acetylation [25] play opposing roles. More importantly, the pathophysiological effects of SERCA2a PTM and the feasibility of the compound-based treatment have been demonstrated in animal studies, making PTM an attractive target. SERCA2a activity and turnover need to be tightly and dynamically regulated in a complex pathological setting such as HF. Thus, modulating PTM to enhance SERCA2a activity or protect its activity along with restoration of protein expression would be one of the ideal strategies for treating HF. Further research is needed to better understand SERCA2a PTM’s dynamics, cross-talking, and regulatory signals during HF progression. It is also necessary to evaluate synergies using SERCA2a gene therapy with PTM target compounds such as SUMOylation activators, Sirt1 activators, and p300 inhibitors. 

Previously, we provided evidence that p300 is directly involved in SERCA2a acetylation. In this study, we identified novel acetylated lysine sites of SERCA2a, which is regulated by p300, and demonstrated that acetylation at K514 is critical for regulating SERCA2a activity in cardiomyocytes. Based on these findings, the regulatory mechanism of SERCA2a acetylation on cardiac function via p300 suggests that it may provide a new target of SERCA2a for the potential treatment of HF.

## 4. Materials and Methos

### 4.1. Adenovirus Generation

Adenoviruses for β-gal, wild-type SERCA2a, and K514Q mutant forms of SERCA2a were generated using the pAdEasy XL adenoviral vector system (Agilent technology, Santa Clara, CA, USA) according to the manufacturer’s protocols.

### 4.2. Calcium-Dependent ATPase Activity Assay

Calcium-dependent ATPase activities of the co-reconstituted proteoliposomes were measured by a coupledenzyme assay. The coupled enzyme assay reagents had the highest purity available (Sigma-Aldrich, Munich, Germany). A minimum of three independent reconstitutions and activity assays were performed for each SERCA2a mutant, and the calcium-dependent ATPase activity was measured over a range of calcium concentrations (0.1 to 10 µM) for each assay. The KCa (calcium concentration at the half-maximal activity), the Vmax (maximal activity), and the nH (Hill coefficient) were calculated based on non-linear least-squares fitting of the activity data to the Hill equation using Sigma Plot software. Errors were calculated as the standard error of the mean for a minimum of three independent measurements. Comparison of KCa, Vmax, and nH parameters was carried out using ANOVA (between-subjects, one-way analysis of variance) followed by the Holm-Sidak test for pairwise comparisons (Sigma Plot).

### 4.3. Cardiomyocyte Isolation and Physiology

Aged 8–10 weeks (weight, 25–30 g), Male C57BL/6J mice were purchased from Jackson Laboratories. Conditional cardiomyocyte-specific *Serca2* knockout mouse has been previously described [26]. Cardiomyocytes were isolated from mouse hearts using the Langendorff method previously described [45]. The culture medium was changed at 4 h after plating, and adenoviruses encoding either control or different SERCA2a constructs were added. As indicators of cardiac functions, intracellular calcium and cell shortening measurements were performed. In brief, the cardiomyocytes were loaded with the calcium indicator Fura-2 and stimulated at different frequencies (0.1–3.0 Hz) using an external stimulator, while a dual excitation spectrofluorometer (IonOptix) was used to record fluorescence emissions used to calculate intracellular calcium concentration. Cell shortening was measured using a video microscopy motion detector system.

### 4.4. Human Heart Samples

Human left ventricular (LV) tissues were obtained from de-identified, failing hearts (with diagnosed cardiomyopathy) at the time of cardiac transplantation or at the time of LV assist device implantation. Non-failing hearts were obtained post-mortem from individuals with no cardiac disease through the National Disease Research Interchange. The Mount Sinai, Institutional Review Board, approved the procurement of the human tissue samples, and all usage of the tissue was performed based on Mount Sinai–approved guidelines. The control donors, including six males and three females, had a median age of 60. The heart failure patients, including five males and two females, had a median age of 62.

### 4.5. Immunoblotting

Heart tissues were homogenized in RIPA buffer (Boston Bioproducts Inc, Milford, MA, USA) with a protease inhibitor cocktail (Roche, Basel, Switzerland) and phosphatase inhibitor cocktail (Sigma). Protein homogenates were separated on an SDS-PAGE gel and transferred to a nitrocellulose membrane (Bio-Rad Laboratories, Hercules, CA, USA). After 1 h blocking with 5% non-fat milk, the membrane was incubated with antibodies against SERCA2a (custom antibody from 21st Century Biochemicals), Acetyl-lysine (Cell Signaling Technology, Leiden, The Netherlands), p300 (Abcam, Cambridge, UK), PCAF (Cell Signaling), GCN5 (Cell Signaling), Flag (Sigma), HA (Cell Signaling), or GAPDH (Sigma) antibodies. Subsequently, the membrane was incubated with HRP-conjugated secondary antibodies (Sigma) and developed using a chemiluminescent substrate (Pierce, ThermoFisher Scientific, Carlsbad, CA, USA).

### 4.6. Immunoprecipitation

Cardiac tissues (4 mg) or cells (500 μg) were washed twice with 1 X PBS and lysed with RIPA buffer containing protease inhibitor mixture and phosphatase inhibitor cocktail. The extracts were prepared using FastPrep Lysing Matrix tubes (MP Biomedicals, ThermoFisher Scientific, Carlsbad, CA, USA) and centrifuged at 17,000× *g* for 15 min at 4 °C. Equal amounts of extracts were incubated with anti-acetyl-lysine agarose beads conjugated to anti-acetyl-lysine (Immunechem Pharmaceuticals, Burnaby, BC, Canada), Flag, SERCA2a, Flag, Myc (Cell Signaling), or HA antibodies at 4 °C on a rotation wheel overnight. The lysate was then incubated with G or A Sepharose beads (Amersham, GE Healthcare, Sweden) for 2 h. The resulting protein complex was washed five times with a lysis buffer and separated by SDS-PAGE and followed by immunoblotting with specific primary antibodies.

### 4.7. In Vitro Acetylation Assay

Purified full-length acetyltransferase proteins (p300 and PCAF) were purchased from Active Motif. SERCA2a was purified by affinity chromatography from porcine hearts. Purified SERCA2a was incubated with or without recombinant human p300 in 20 μL of reaction buffer (50 mM Tris-HCl at pH 8.0, 10 mM Sodium Butyrate, 0.8 mM EDTA, 10% Glycerol, and 1 mM DTT) containing 10 μM Acetyl-CoA (Sigma) at 37 °C for 1 h and then placed on ice for 15 min. Sample buffer was added to the reaction mix, followed by analysis by immunoblotting using an anti-acetyl-lysine antibody.

### 4.8. Mass Spectrometry Analysis

Purified SERCA2a untreated and treated with recombinant p300 protein was separated on SDS-PAGE. The band corresponding to SERCA2a (approximately 110 kDa) was excised and submitted for MS/MS analysis. The MS/MS spectra obtained from the LC/LC-ESI-MS/MS analyses were used to search the mouse International Protein Index database (ver. 3.78) using Bioworks software (ver. 3.2). Oxidation of methionine and acetylation of lysine were assigned in SEQUEST searches as variable modifications (methionine +16, lysine +42), with a maximum of three modifications allowed per peptide (the maximum number of modifications per type was five) and a maximum of two missed cleavage sites for trypsin digestion.

### 4.9. Statistical Analysis

All molecular and biochemical data are reported as the mean ± SD. Data were analyzed in either Excel or GraphPad Prism software (ver. 9) (GraphPad Software, La Jolla, CA, USA). Statistical differences between the two groups were tested with a two-tailed Student’s *t*-test. Statistical differences between more than two groups were analyzed with one-way ANOVA tests followed by Tukey’s multiple comparison tests. *p* < 0.05 was considered statistically significant.

## Figures and Tables

**Figure 1 ijms-24-03502-f001:**
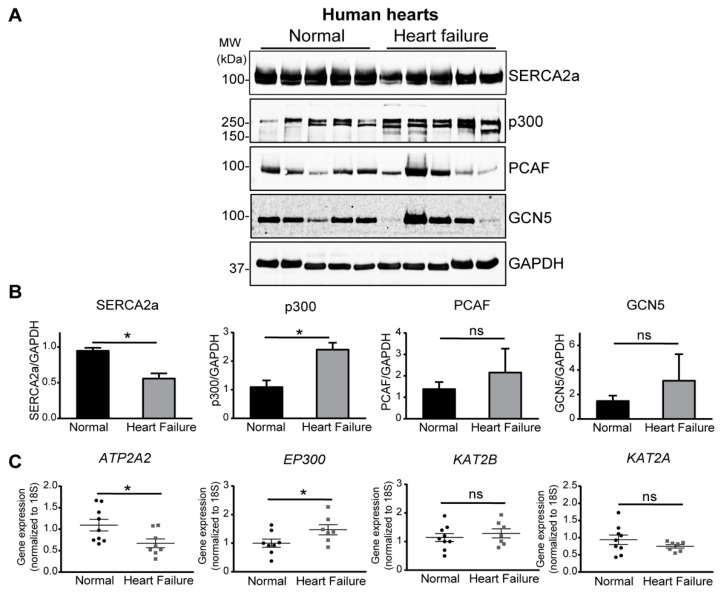
p300 is increased in human heart failure. (**A**) Heart biopsy samples from patients with heart failure and normal controls were homogenized and immunoblotted with the indicated antibodies. Representative immunoblots of SERCA2a and key acetyltransferases (p300, PCAF, GCN5) in heart biopsies from normal and heart failure individuals are shown. (**B**) Quantification plots of immunoblot data. (**C**) mRNA levels of SERCA2a and key acetyltransferases (p300, KAT2A, KAT2B) in heart biopsies from normal and heart failure individuals. Data are represented as the mean ± SD. Donors, *n* = 9; failing patients, *n* = 8. ns, not significant; * *p* < 0.05 vs. normal by unpaired *t*-test.

**Figure 2 ijms-24-03502-f002:**
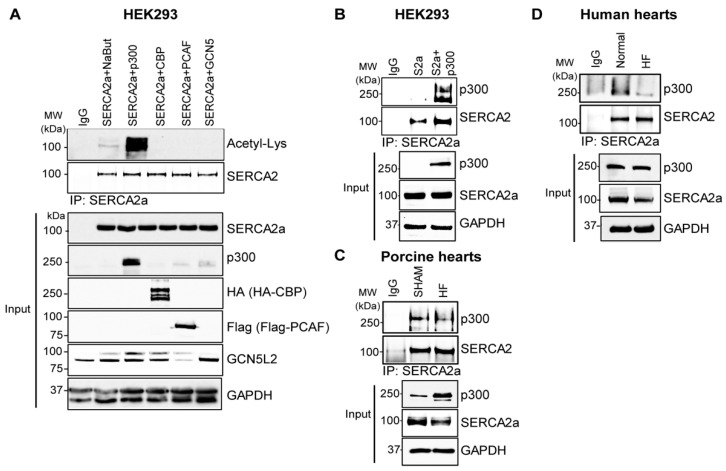
p300 interacts with and acetylates SERCA2a. (**A**) HEK293 cells were transfected with SERCA2a alone or SERCA2a in combination with different acetyltransferases (p300, CBP, PCAF, GCN5). Immunoprecipitated with anti-SERCA2a antibody was conducted with lysates of these HEK293 cells and probed with anti-acetyl-lysine antibody. (**B**) HEK293 cells were transfected with SERCA2a alone or SERCA2a in combination with p300. Immunoprecipitated with anti-SERCA2a antibody was conducted with lysates of these HEK293 cells and probed with anti-P300 antibody. (**C**) Immunoprecipitation of SERCA2a from sham-operated and myocardial infarction-induced heart failure pig hearts showing the interaction between SERCA2a and p300. (**D**) Immunoprecipitation of SERCA2a from normal and heart failure human patient hearts showing the interaction between SERCA2a and p300.

**Figure 3 ijms-24-03502-f003:**
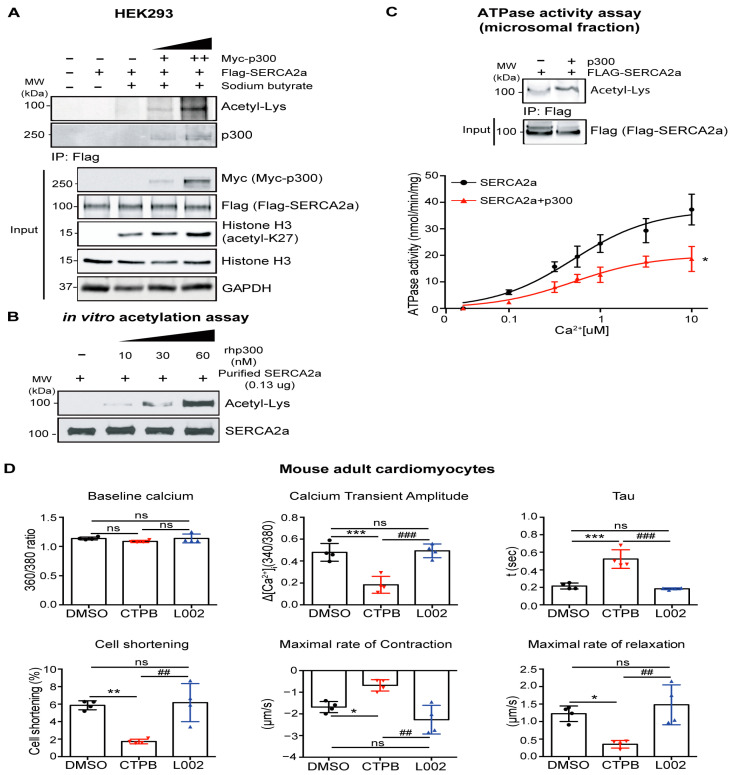
p300 acetylates SERCA2a and diminishes its activity. (**A**) HEK-293 cells were transfected with SERCA2a alone or SERCA2a in combination with different dosages of p300 plasmid. Anti-Flag agarose beads were incubated with lysates of these HEK293 cells. Sodium butyrate was treated to inhibit the deacetylation of SERCA2a. Immunoprecipitants were probed with anti-acetyl-lysine or anti-p300 antibodies. (**B**) SERCA2a protein was purified from normal pig hearts and incubated with recombinant human p300 (rhp300). The in vitro acetylation was probed with anti-acetyl-lysine and anti-SERCA2a antibodies. (**C**) Calcium-dependent ATPase activity was measured using microsomal fractions isolated from HEK-293 cells expressing SERCA2a alone (black line) or SERCA2a and p300 (red line). Data are represented as the mean ± SD of *n* = 3. * *p* < 0.05 vs. SERCA2a alone by paired *t*-test. (**D**) Mouse adult cardiomyocytes (ACMs) were isolated from male C57BL/6J mice and treated with 5 μM CTPB (p300 activator) or 5 μM L002 (p300 inhibitor). 24 h after treatment with CTPB or L002, the contractile response of ACMs was assessed by calcium transient amplitude, decay time constant (tau), peak shortening, maximal rate of contraction, and maximal rate of relaxation using a video-based edge-detection system (IonOptix, Inc, Milton, MA, USA). Fifteen cardiomyocytes were measured per mouse, *n* = 4. Graphs show mean ± SD, with each data point represented by a mean average of 15 cardiomyocytes isolated from one heart sample. ns, not significant; * *p* < 0.05; ** *p* < 0.01; *** *p* < 0.001 vs. DMSO-treated ACMs and ## *p* < 0.01; ### *p* < 0.001 vs. CTPB-treated ACMs by one-way ANOVA.

**Figure 4 ijms-24-03502-f004:**
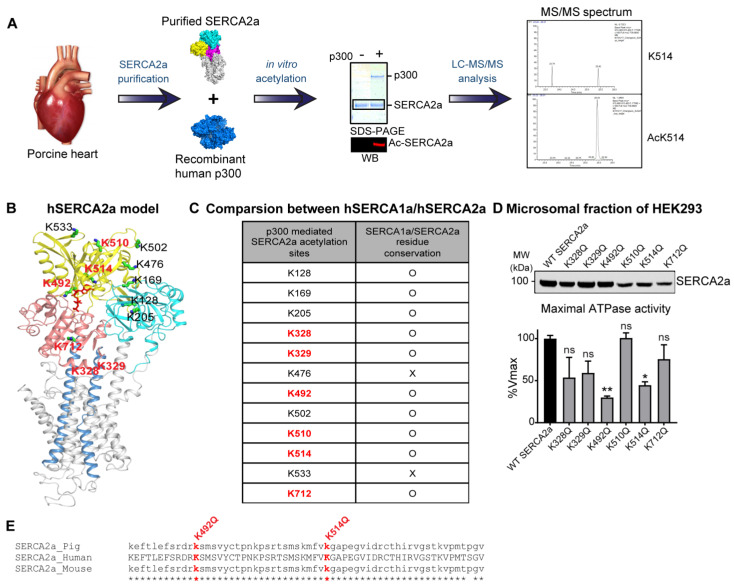
Identification and functional characterization of p300-mediated acetylation sites in SERCA2a. (**A**) Schematic diagram depicting the identification of lysine residues acetylated by p300. SERCA2a protein was purified by affinity chromatography from normal porcine hearts and acetylated in vitro by purified recombinant human p300 acetyltransferase. Acetylation was verified by immunoblot. SERCA2a protein bands were cut out of a Coomassie-stained SDS-PAGE gel, and MS/MS analysis was performed. (**B**) Cartoon representation of SERCA2a based on the crystal structure of SERCA1a (PDB: 1T5S). The phosphorylation domain (salmon), nucleotide-binding domain (yellow), actuator domain (cyan), and transmembrane domain (grey) are shown, respectively. The putative acetylated lysine residues in SERCA2a identified by MS/MS analysis were presented. Six acetylated lysine sites (e.g., lysine 328, 329, 492, 510, 514, and 712) that appeared in all MS analyses were shown in red. (**C**) The table has summarized the acetylation site conservation between SERCA2a and SERCA1a. (**D**) Summary of Vmax obtained from calcium-dependent ATPase activity assay of microsomes isolated from HEK-293 cells expressing wild-type SERCA2a or acetyl-mimicking mutants of SERCA2a. Data are represented as the mean ± SEM of *n* = 3~5 experiments. ns, not significant; * *p* < 0.05; ** *p* < 0.01 vs. WT SERCA2a by one-way ANOVA. A representative immunoblot of the isolated microsomes containing the different SERCA2a constructs is shown above. (**E**) Protein sequence alignment of human, porcine, and mouse SERCA2a show the perfect conservation of Lys492 and Lys514 acetylation sites. Below the protein sequences is a key denoting conserved sequence (*).

**Figure 5 ijms-24-03502-f005:**
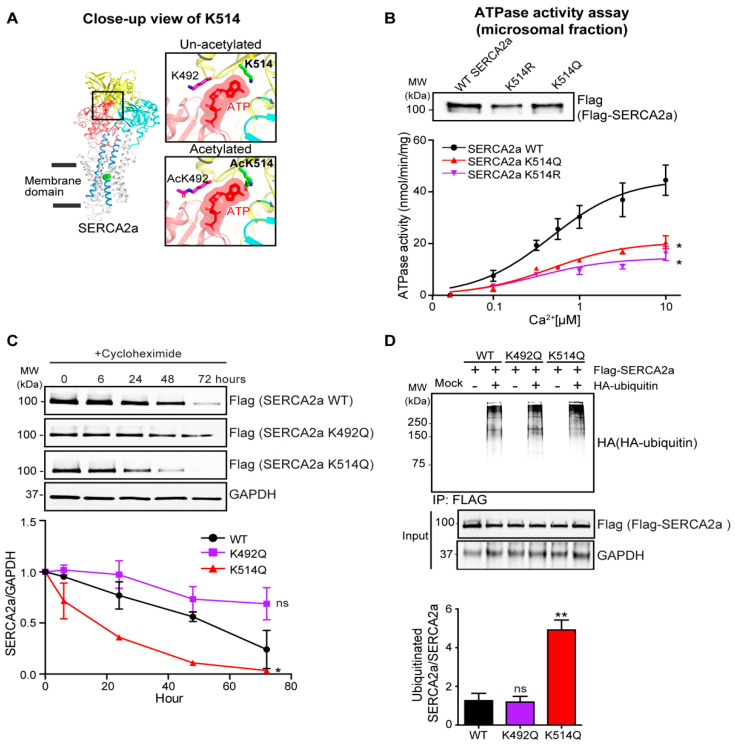
Acetylation on Lys514 diminishes the activity and stability of SERCA2a. (**A**) A structural model of SERCA2a showing Lys492 and Lys514 acetylation sites in SERCA2a. Zoom-in panels show un-acetylated and acetylated forms of the two acetylation sites in the ATP binding pocket of SERCA2a. (**B**) Calcium-dependent ATPase activity of wild-type and acetylation-mimicking mutant forms of SERCA2a. A representative immunoblot of the isolated microsomes containing the different SERCA2a constructs is shown above. Data are represented as the mean ± SD of *n* = 3 experiments. * *p* < 0.05 vs. wild-type (WT) SERCA2a by paired *t*-test. (**C**) Immunoblot is showing a time-dependent experiment of WT, K492Q, and K514Q SERCA2a protein levels following cycloheximide treatment. The quantification blot is shown below. Graphs show means ± SD of *n* = 3 experiments. ns, not significant; * *p* < 0.05 vs. WT SERCA2a by paired *t*-test. (**D**) Immunoblots are showing in vitro ubiquitination assays of WT, K492Q, and K514Q SERCA2a. The quantification blot is shown below. Graphs show means ± SD of *n* = 3 experiments. ns, not significant; ** *p* < 0.01 vs. WT SERCA2a by one-way ANOVA.

**Figure 6 ijms-24-03502-f006:**
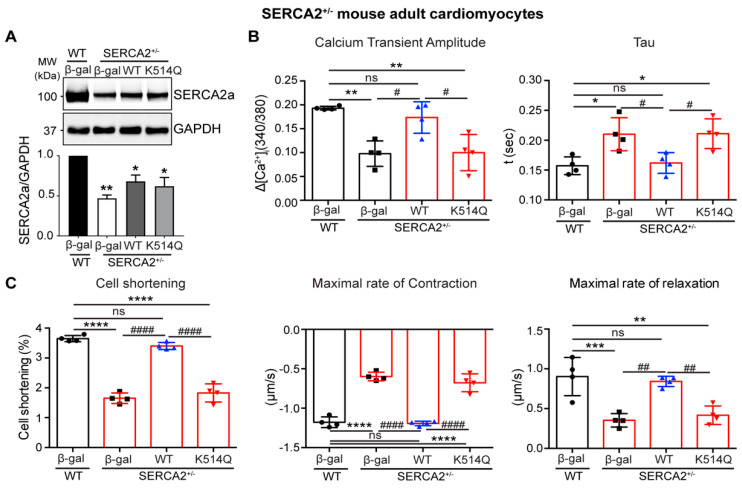
Acetylation on Lys514 of SERCA2a impairs cardiomyocyte function. (**A**) Immunoblot shows levels of SERCA2a in cardiomyocytes isolated from wild-type (WT) and SERCA2 heterozygous knock-down mice (SERCA2^+/−^) infected with respective adenoviruses. (**B**) Calcium transient summary showing calcium transient amplitude and tau values. (**C**) Adult cardiomyocyte contractility data show peak shortening, maximal rate of relaxation, and maximal rate of contraction values. The function of ACMs was assessed using a video-based edge-detection system (IonOptix, Inc, Milton, MA) 24 h after infection of ACMs with adenovirus (50 MOI of each virus). *n* = 4. Graphs show means ± SD, with each data point represented by a mean average of 15 ACMs isolated from one heart sample. ns, not significant; * *p* < 0.05; ** *p* < 0.01; *** *p* < 0.001; **** *p* < 0.0001 vs. WT + β-gal and # *p* < 0.05; ## *p* < 0.01; #### *p* < 0.0001 vs. SERCA2^+/−^ + WT SERCA2a by one-way ANOVA.

**Table 1 ijms-24-03502-t001:** Summary of acetylation sites in SERCA2a identified by mass spectrometry analysis.

Acetylated Lysine	Peptide Sequence	Detected Monoisotopic Mass (m/z)	Actual Mass (m/z)	Delta Da	*n*
K128	EYEPEMG**K***VYR	729.83	1457.65	0.0024	3
K169	LTSI**K***STTLR	581.35	1160.68	0.0018	3
K205	**K***NMLFSGTNIAAGK	755.39	1508.77	0.0027	3
K328	MA**K***KNAIVR	565.82	1129.63	0.00082	2
K329	MAK**K***NAIVR	565.82	1129.63	0.00082	2
K476	ANACNSVI**K***QLMK	767.89	1533.77	0.0031	2
K492	**K***SMSVYCTPNKPSR	856.91	1711.80	0.0027	3
K502	KSMSVYCTPN**K***PSR	856.91	1711.80	0.0027	2
K510	TSMS**K***MFVK	566.77	1131.53	0.0019	4
K514	MFV**K***GAPEGVIDR	738.88	1475.75	0.0042	4
K533	VGST**K***VPMTPGVK	679.87	1357.73	−0.00061	3
K541	VPMTPGV**K***QK	571.82	1141.62	0.0014	2
K712	**K***SEIGIAMGSGTAVAK	789.42	1576.82	0.0029	4

The acetylated K is bolded and noted with *.

**Table 2 ijms-24-03502-t002:** Kinetic parameters for wild-type and acetylation mimicking mutants of SERCA2a.

SERCA2a Form	V_max_ (%)		K_Ca_ (µM)		n_H_		N
	AVR	SEM	AVR	SEM	AVR	SEM	
WT	100.0	3.7	0.45	0.0	1.3	0.2	5
K328Q	53.7	24	0.47	0.0	1.5	0.1	3
K329Q	59.2	14	0.44	0.1	1.8	0.3	3
K492Q	30.0	1.7	0.25	0.0	1.8	0.4	3
K510Q	100.7	6.0	0.43	0.1	1.3	0.3	3
K514Q	44.6	4.0	0.48	0.1	1.4	0.4	3
K712Q	75.6	17	0.40	0.0	1.3	0.0	3

Abbreviations: V_max_, maximum reaction velocity; K_Ca_, calcium concentration at half-maximal activity; n_H_, Hill coefficient; N, number of animals.

## Data Availability

Not applicable.

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
