# Peer review of "Identification and Characterization of p300-Mediated Lysine Residues in Cardiac SERCA2a"

_ijms, 2023, doi:10.3390/ijms24043502_

Round 1

Reviewer 1 Report

Gorski et al. examined the role of lysine acetylation, a putative post-translational protein modification (PTM) in the reduction of SERCA2a activity during heart failure (HF). They found that SERCA2a acetylation is higher in failing than in non-failing human hearts. Moreover, they showed that acetyltransferase p300 can be responsible for this effect, and SERCA2a Lys514 (K514) was highlighted as a significant regulator of SERCA2a activity. Collectively, results of this investigation illustrates p300 mediated acetylation as a potential mechanism contributing to the malfunction of SERCA2a and cardiac systolic dysfunction in HF. The applied methods represent high standards, the results are convincing, the paper reads well, and the conclusions are sound.

Minor:

1/ The molecular interaction between P300 and SERCA2a is considered as crucial in cardiac dysfunction here. How far can you exclude the involvement of P300 mediated acetylation of proteins other than SERCA2a in HF? Potential acetylation of proteins of myofilaments and Ca2+ cycling can be particularly interesting from this viewpoint. Please, discuss!

2/ The activity of SERCA2a is regulated by phospholamban (PLN) to a great degree. Are you aware of an interaction between SERCA2a acetylation and PLN effects on SERCA2a activity? Please, discuss!

3/ Would it be possible to judge (based on your present data and literature information) how far SERCA2a acetylation can be accounted for the diminished SERCA2a function relative to other putative molecular mechanisms (e.g. changes in SERCA2a/phospholamban expression/phosphorylation, etc.) during HF? Please, discuss!

4/ In Figures 3 and 6 (in panels, legends) ad in the related section in the Results section (page 6, top) you refer to “calcium amplitude”. I believe it is more appropriate to use the “calcium transient amplitude” expression. In addition, kindly consider unification of the labelling of panels in figures (i.e. the use of capital letters in titles, font sizes and font types).

Author Response

We would like to thank all reviewers for taking the time and effort necessary to review the manuscript. We sincerely appreciate all valuable comments and suggestions which helped us to improve the quality of the manuscript. We believe all the comments have been fully addressed. Please see below (highlighted in blue) our detailed response to comments.

REVIEWER 1

Gorski et al. examined the role of lysine acetylation, a putative post-translational protein modification (PTM) in the reduction of SERCA2a activity during heart failure (HF). They found that SERCA2a acetylation is higher in failing than in non-failing human hearts. Moreover, they showed that acetyltransferase p300 can be responsible for this effect, and SERCA2a Lys514 (K514) was highlighted as a significant regulator of SERCA2a activity. Collectively, results of this investigation illustrates p300 mediated acetylation as a potential mechanism contributing to the malfunction of SERCA2a and cardiac systolic dysfunction in HF. The applied methods represent high standards, the results are convincing, the paper reads well, and the conclusions are sound.

Minor:

  1. The molecular interaction between P300 and SERCA2a is considered as crucial in cardiac dysfunction here. How far can you exclude the involvement of P300 mediated acetylation of proteins other than SERCA2a in HF? Potential acetylation of proteins of myofilaments and Ca2+ cycling can be particularly interesting from this viewpoint. Please, discuss!

Thank you for your insightful comments.

To date, studies on acetylation in heart disease largely focused on the histone acetylation to regulate gene expression. However, several studies have reported the involvement of non-histone acetylation in the pathophysiological progress of cardiac hypertrophy and arrhythmia. For example, abnormal acetylation of mitochondrial antioxidant enzyme SOD2, Ca2+ handling protein SERCA2a, and gap junction protein connexin 43 cause excessive oxidative stress, metabolic alternation, impaired calcium homeostasis, resulting in cardiac dysfunction.

This study focuses on the role of p300-mediated acetylation in SERCA2a biology. However, acetyltransferase p300 may affect other molecules involved in heart functions, including other Ca2+ handling proteins and contractile machinery. In addition, the in silico acetylation site analysis suggests that the RyR2 Ca2+ channel protein, sodium-calcium exchanger, and troponin I contain many acetylation sites (data not shown). Also, some of these proteins undergo other PTMs, such as phosphorylation, thus it cannot be ruled out that they could be direct or indirect targets of p300.

Further investigation, such as identifying p300 interactome in normal and failing hearts, will help us understand the relationship between cardiac function and acetylation of key molecules in terms of molecular physiology.

Part of this discussion is now added to the revised manuscript.

  1. The activity of SERCA2a is regulated by phospholamban (PLN) to a great degree. Are you aware of an interaction between SERCA2a acetylation and PLN effects on SERCA2a activity? Please, discuss!

Thank you very much for your insightful questions. 

In this study, we identified 12 lysine residues on SERCA2a potentially acetylated by p300 and studied their relationship with the function of SERCA2a, focusing on lysine 514. Interestingly, we identified two acetylated lysine residues such as K328 and K329, in the putative PLN binding pocket of SERCA2a. K328 has already been reported to cross-link with Q23 and K27 of PLN [Morita T et al, 2008; Akin & Jones, 2012]. Through IP experiments in overexpression systems, it was observed that the acetylation-mimic mutant of K328 SERCA2a increases its affinity for PLN by ~2.5-fold compared to wild-type SERCA2a (Provided Figure 1, data not shown). In-depth studies will be needed to clarify the role of acetylation in the SERCA2a-PLN complex.

Part of this discussion is now added to the revised manuscript.

ProFig.1: Effects of acetylation on interaction between SERCA2a and PLN.

Wild-type or single mutant was transfected with PLN into HEK293 cells. Immunoprecipitation products were subjected to SDS-PAGE and Western blotting.  The bands of PLN proteins were normalized with SERCA2a bands.

  1. Would it be possible to judge (based on your present data and literature information) how far SERCA2a acetylation can be accounted for the diminished SERCA2a function relative to other putative molecular mechanisms (e.g. changes in SERCA2a/phospholamban expression/phosphorylation, etc.) during HF? Please, discuss!

Thank you very much for your insightful comments.

SERCA2a is a promising molecular target for HF therapy. PLN is a well-known regulatory protein of SERCA2a, and its reduced phosphorylation contributes to impaired SERCA2a function in failing hearts. Many efforts to increase PLN phosphorylation have led to the development of the continuously active form of protein phosphatase inhibitor I1 (I1c). I1c gene therapy recently initiated a Phase 1 clinical study (NCT04179643). A recent study revealed another protein DWORF which functions as an activator of SERCA2a via replacing PLN [Makarewich et al, 2018].

Several groups, including us, have shown that PTMs are important regulatory mechanisms of SERCA2a activity. SUMOylation and glutathionylation [Adachi et al, 2004] function as SERCA2a stimulators, whereas oxidation [Vangheluwe et al, 2005] and acetylation play opposing roles. More importantly, the pathophysiological effects of SERCA2a PTM and the feasibility of the compound-based treatment have been demonstrated in animal studies, making PTM an attractive target. SERCA2a activity and turnover need to be tightly and dynamically regulated in a complex pathological setting such as HF. Thus, modulating PTM to enhance SERCA2a activity or protect its activity along with restoration of protein expression would be one of the ideal strategies for treating HF. Further research is needed to better understand SERCA2a PTM’s dynamics, cross-talking, and regulatory signals during HF progression. It is also necessary to evaluate synergies using SERCA2a gene therapy with PTM target compounds such as SUMOylation activators, Sirt1 activators, and p300 inhibitors.

This discussion is now added to the revised manuscript.

  1. In Figures 3 and 6 (in panels, legends) ad in the related section in the Results section (page 6, top) you refer to “calcium amplitude”. I believe it is more appropriate to use the “calcium transient amplitude” expression. In addition, kindly consider unification of the labelling of panels in figures (i.e. the use of capital letters in titles, font sizes and font types).

As per reviewer’s suggestion, in the revised manuscript, we replaced calcium transient amplitude instead with calcium amplitude in the related section of Figures 3 and 6 (in panels, legends). Also, we revised to unify the labeling of panels in the figures.

Reviewer 2 Report

This is a compelling work including an elegant study design which has been soundly executed and analyzed. The work of great significance to the cardiology research and PTM research communities and is certain to be of interest to the readership of this journal. This referee finds no issues with the design, presentation, interpretation or conclusions that would delay immediate acceptance. I applaud the authors for a thorough, highly valuable work. 

Author Response

We sincerely appreciate your comments and supportive praise for providing motivation and drive for future research.

Reviewer 3 Report

In this manuscript, authors claimed that (1) p300 expression is induced in human heart failure, (2) p300 interacts with SERCA2a and acetylated multiple lysine residues within them, (3) acetylated SERCA2a has lower ATPase activity, (4) among these acetylation sites, the acetylation of K514 significantly affects SERCA2a protein stability and ATPase activity in mouse adult cardiomyocytes. The manuscript was well written, and the authors clearly showed the function of SERCA2a acetylation using multifaceted approaches such as in vivo human heart tissue samples, cultured adult mouse cardiomyocytes, in vitro biochemical assay, and crystal structure modeling. The authors have previously found SUMO-1 modification of SERCA2a, it is very interesting that multiple types of post-translational modification regulate SERCA2a. Nevertheless, I have some concerns as follows.

Major concern;

1.      P300 acetyltransferase is a well-known chromatin modifier such as histone acetylation and is ubiquitously expressed in almost organs. The authors should discuss the reason why p300 increased in failing hearts.

2.      Under physiological conditions of the heart, is SERCA2a acetylated? If so, how its acetylation is regulated? In failing hearts, does de-regulated p300 acetylate SERCA2a?

3.      The authors previously showed that SERCA2a was modified with SUMO-1 at K480 and K585. What is the functional relationship between SUMOylation and acetylation of SERCA2a?

4.      In Figure3D, the authors showed that small compounds CTPB and L002 modulated Ca++ handling in adult cardiomyocytes. The authors should examine whether these compounds actually modulate the acetylation status of SERCA2a in these cultured cardiomyocytes.

Minor concerns;

1.      In the graph of ATPase activity in Figure 3C, the unit of the x-axis is incorrect “uM”. The authors should correct this.

2.      In Figure 4D, the authors should show the loading control of microsomal fraction to demonstrate the protein instability of SERCA2a mutants.

3. On page 7, second line from the bottom, the authors described “K328Q, K492Q, and K514Q SERCA2a exhibited significantly decreased ATPase activities”. However, In Figure 4D, the “ns” is stated above the bar graph of K328Q. Which is correct?

Author Response

RESPONSE TO REVIEWERS

We would like to thank all reviewers for taking the time and effort necessary to review the manuscript. We sincerely appreciate all valuable comments and suggestions which helped us to improve the quality of the manuscript. We believe all the comments have been fully addressed. Please see below (highlighted in blue) our detailed response to comments.

REVIEWER 3

In this manuscript, authors claimed that (1) p300 expression is induced in human heart failure, (2) p300 interacts with SERCA2a and acetylated multiple lysine residues within them, (3) acetylated SERCA2a has lower ATPase activity, (4) among these acetylation sites, the acetylation of K514 significantly affects SERCA2a protein stability and ATPase activity in mouse adult cardiomyocytes. The manuscript was well written, and the authors clearly showed the function of SERCA2a acetylation using multifaceted approaches such as in vivo human heart tissue samples, cultured adult mouse cardiomyocytes, in vitro biochemical assay, and crystal structure modeling. The authors have previously found SUMO-1 modification of SERCA2a, it is very interesting that multiple types of post-translational modification regulate SERCA2a. Nevertheless, I have some concerns as follows.

Major concern;

  1. P300 acetyltransferase is a well-known chromatin modifier such as histone acetylation and is ubiquitously expressed in almost organs. The authors should discuss the reason why p300 increased in failing hearts.

It was reported that dysregulated p300 activity and expression are associated with various of human diseases. Significant elevation of p300 expression is found in skin and lung fibrosis, kidney disease, and liver disease.

In the heart, p300 has been found to play an important role in the pathological processes of hypertrophy and heart failure. Its expression is increased by cardiac stresses such as hypertension, diabetes, and vascular injury and controls cardiac gene expression. p300 is known to be a key driver of cardiac hypertrophic response by activating transcription factors involved in cardiac myocyte growth, such as MEF2 and GATA4. p300 is also an essential epigenetic regulator of fibrogenesis. It has been demonstrated that p300 stimulates the synthesis of type I collagen, a key matrix protein that contributes to fibrogenesis, and requires acetyltransferase activity [Ghosh et al, 2000]. The importance of p300 function in heart failure was further validated in animal models of hypertension and heart failure. Pharmacological inhibition of acetyltransferase activity of p300 improved structural abnormalities and cardiac dysfunction induced by cardiac injury [Morimoto et al, 2008; Rai et al, 2017]. However, upstream regulators and detailed control mechanisms of p300 expression in failing hearts are uncertain.

Part of this discussion is now added to the revised manuscript.

  1. Under physiological conditions of the heart, is SERCA2a acetylated? If so, how its acetylation is regulated? In failing hearts, does de-regulated p300 acetylate SERCA2a?

It is thought that SERCA2a is acetylated under physiological conditions. Our previous studies clearly indicated that SERCA2a is acetylated in normal hearts, and this acetylation is more prominent in the setting of heart failure. We have further detected endogenous interaction between SERCA2a and p300 in normal heart tissues. Some of the p300-dependent acetylation sites, including lysine 514, have also been reported in normal rat, pig, and human hearts.

Importantly two critical enzymes for the acetylation switch of SERCA2a, such as p300 acetyltransferase and Sirt1 deacetylase, are both dysregulated in failing hearts. It is reported that p300 itself is also a target of Sirt1. More research is needed to understand the regulatory mechanisms of expression and activity of both enzymes in the context of heart failure.

  1. The authors previously showed that SERCA2a was modified with SUMO-1 at K480 and K585. What is the functional relationship between SUMOylation and acetylation of SERCA2a?

Thank you for the insightful questions.

This manuscript focuses on confirming that p300 interacts with and acetylates SERCA2a. We have compelling data indicating that SUMOylation affects acetylation at some, but not all, lysine residues of SERCA2a (please see below) from the previous studies. Since the two lysine residues (K480 and K585) of SERCA2a subject to SUMOylation are not acetylated, there should be no direct competition between SUMOylation and acetylation. Instead, these two modifications appear to be regulated through independent mechanisms but in opposite directions. Yet, SUMOylation, due to the relatively large size of SUMO1, may sterically hinder the acetylation at near lysine residues.

We provide here two “Provided Figures” to be viewed only by reviewers. In Provided Figure 2, SERCA2a acetylation was increased under pressure overload, which was significantly prevented by AAV-mediated SUMO1 overexpression. In Provided Figure 3, on the contrary, SERCA2a acetylation was increased considerably when the SUMO1 level was reduced by shRNA transfer.

Taken together, These data seem to be cross-talks between acetylation and SUMOylation of SERCA2a. However, we think this issue is beyond the scope of this manuscript.

  1. In Figure3D, the authors showed that small compounds CTPB and L002 modulated Ca++ handling in adult cardiomyocytes. The authors should examine whether these compounds actually modulate the acetylation status of SERCA2a in these cultured cardiomyocytes.

We fully agree with the reviewer's point about whether these compounds regulate the acetylated state of SERCA2a  in the isolated cardiomyocytes. It was confirmed that both compounds modulate the acetylation status of SERCA2a in ACMs as well as in vitro assays (Figure S1). SERCA2a acetylation was increased by treatment with CTPB, while its acetylation was remarkably decreased after treatment with L002. Either CTPB or L002 treatment did not alter SERCA2a protein expression levels in ACMs.

The data with part of description has now been added as a supplement to the revised manuscript.

Minor concerns;

  1. In the graph of ATPase activity in Figure 3C, the unit of the x-axis is incorrect “uM”. The authors should correct this.

In our calcium-dependent ATPase activity assay, we employed a coupled-enzyme assay. Since the calcium-dependent ATPase activity was measured over a range of calcium concentrations (0.1 to 10 μM) for each assay, the unit of the x-axis is correct “μM”. 

  1. In Figure 4D, the authors should show the loading control of microsomal fraction to demonstrate the protein instability of SERCA2a mutants.

In accordance with the reviewer’s suggestion, we provided loading control image (Provided Figure 4) that matches Figure 4D. Since there is no specific internal loading control of microsomal fraction, the normalization of SERCA2a level was standardized with the whole protein band obtained by Coomassie brilliant blue G-250 staining after loading the same amount. It should be remembered that Figure 4D is a result of explaining the activity of the SERCA2a mutation rather than dealing with SERCA2a protein instability. Although the SERCA2a mutation appeared differently in Figure 4D, the protein activity presented a normalized value for each protein amount.

Also, we already showed that the protein instability of K514Q SERCA2a mutants is decreased compared with wild-type SERCA2a. To estimate accurate protein instability of SERCA2a mutants, we measured the SERCA2a mutants' stability after cycloheximide treatment with the expression of the SERCA2a mutation (Figure 5 C).

  1. On page 7, second line from the bottom, the authors described “K328Q, K492Q, and K514Q SERCA2a exhibited significantly decreased ATPase activities”. However, In Figure 4D, the “ns” is stated above the bar graph of K328Q. Which is correct?

Thanks for pointing that out, and we fixed the typo on page 7. In addition, a new sentence was added to the revised manuscript: "K328Q, K492Q, and K514Q SERCA2a exhibit > 45% reduced ATPase activity compared to WT SERCA2a."

Round 2

Reviewer 1 Report

Thank you for your responses, I have no further comments.

Author Response

(The authors gave the same response as above.)

Reviewer 3 Report

The authors have addressed most of the reviewer's concerns, except for one minor point.

1.      In the graph of ATPase activity in Figure 3C, the unit of the x-axis is incorrect “uM”. The authors should correct this.

In our calcium-dependent ATPase activity assay, we employed a coupled-enzyme assay. Since the calcium-dependent ATPase activity was measured over a range of calcium concentrations (0.1 to 10 μM) for each assay, the unit of the x-axis is correct “μM”. 

The reviewer understands that Figure 3C shows a coupled-enzyme assay and that “μM” is correct.

The reviewer pointed out the misspelling of “u”M in the unit of the x-axis of Figure 3C.

Additionally, similar misspellings are also found in Figure 3B (0.13 “u”g) and Figure 5B (Ca2+[“u”M])

Author Response

We appreciate the reviewer’s remark. We clarified the unit of “μM” in Figure 3B, 3C, and Figure 5B in the legend of the Revised Figures.

Round 3

Reviewer 3 Report

The authors have addressed all of the reviewer's concerns.